# PD-1 Expression Promotes Immune Evasion in B-ALL

**DOI:** 10.3390/hematolrep17060061

**Published:** 2025-11-12

**Authors:** Ana Casado-García, Gonzalo García-Aguilera, Julio Pozo, Ninad Oak, Susana Barrena, Belén Ruiz-Corzo, Jaanam Lalchandani, Ana Chamorro-Vera, Ana Castillo-Robleda, Beatriz Soriano, Silvia Alemán-Arteaga, Elena G. Sánchez, Jorge Martínez-Cano, Andrea López-Álvarez de Neyra, Paula Somoza-Cotillas, Oscar Blanco, Susana Riesco, Pablo Prieto-Matos, Francisco Javier García Criado, María Begoña García Cenador, César Cobaleda, Carolina Vicente-Dueñas, Kim E Nichols, Alberto Orfao, Manuel Ramírez-Orellana, Isidro Sánchez-García

**Affiliations:** 1Experimental Therapeutics and Translational Oncology Program, Instituto de Biología Molecular y Celular del Cáncer, CSIC-USAL, Campus M. de Unamuno s/n, 37007 Salamanca, Spain; anacasagar@usal.es (A.C.-G.); belen.ruizcorzo@usal.es (B.R.-C.); alemanarteaga.silvia@usal.es (S.A.-A.); andrea.alvarezdeneyra@usal.es (A.L.-Á.d.N.); paulasomoza@usal.es (P.S.-C.); 2Instituto de Investigación Biomédica de Salamanca (IBSAL), Universidad de Salamanca, Campus Miguel de Unamuno s/n, 37007 Salamanca, Spain; oscarblancomunez@usal.es (O.B.); sriesco@saludcastillayleon.es (S.R.); pprieto@saludcastillayleon.es (P.P.-M.); fjgc@usal.es (F.J.G.C.); mbgc@usal.es (M.B.G.C.);; 3Fundación para la Investigación Biomédica del Hospital Universitario La Princesa, 28006 Madrid, Spain; gonzalo.garcia.externo@salud.madrid.org; 4Servicio de Citometría, Departamento de Medicina, Biomedical Research Networking Centre on Cancer CIBER-CIBERONC (CB16/12/00400), Institute of Health Carlos III, and Instituto de Biología Molecular y Celular del Cáncer (CIC-IBMCC), CSIC/Universidad de Salamanca, 37007 Salamanca, Spain; juliopozo14@usal.es (J.P.); subadelfa@usal.es (S.B.); bsoriano21@usal.es (B.S.); 5Department of Oncology, St. Jude Children’s Research Hospital, 262 Danny Thomas Pl, Memphis, TN 38105, USA; ninad.oak@stjude.org; 6Advance Therapy Unit, Oncohematology Department, Fundación para la Investigación Biomédica del Hospital Infantil Universitario del Niño Jesús, Avenida Menendez y Pelayo, 65 28009 Madrid, Spain; jaanamlalchandani@gmail.com (J.L.); elena.garcia.sanchez@salud.madrid.org (E.G.S.); 7Department of Pediatric Hematology and Oncology, Hospital Infantil Universitario Niño Jesús, Universidad Autónoma de Madrid, Avenida Menendez y Pelayo, 65, 28009 Madrid, Spain; anachamorrov93@gmail.com (A.C.-V.); castillorobleda@hotmail.com (A.C.-R.); 8Immune System Development and Function Unit, Centro de Biología Molecular Severo Ochoa (Consejo Superior de Investigaciones Científicas-Universidad Autónoma de Madrid), 28049 Madrid, Spain; jorge.martinez@cbm.csic.es (J.M.-C.); cesar.cobaleda@csic.es (C.C.); 9Departamento de Anatomía Patológica, Universidad de Salamanca, 37007 Salamanca, Spain; 10Department of Pediatrics, Hospital Universitario de Salamanca, Paseo de San Vicente, 58-182, 37007 Salamanca, Spain; 11Departamento de Cirugía, Universidad de Salamanca, 37007 Salamanca, Spain

**Keywords:** preleukemic cells, leukemia, acute, childhood, immunotherapy, murine models, genetic susceptibility, immune evasion

## Abstract

Background/Objectives: In children developing B-cell acute lymphoblastic leukemia (B-ALL), an immune evasion event takes place where otherwise “silent” preleukemic cells undergo a malignant transformation while escaping immune control, often through unknown mechanisms. Methods and Results: Here, we identify the upregulation of PD-1 expression in preleukemic cells, triggered by *Pax5* inactivation in mice and correlating with the time of conversion to leukemia, as a novel marker that favors leukemia evasion. This increase in PD-1 expression is apparent across diverse molecular B-ALL subtypes, both in mice and humans. PD-1 is not required for B-cell leukemogenesis, but, in the absence of PD-1, tumor cells express NK cell inhibitory receptors, highlighting the necessity for leukemic cells to evade the host’s NK immune response in order to exit the bone marrow. PD-1 expression reduces natural antitumor immune responses, but it sensitizes leukemic cells to immune checkpoint blockade strategies in mice and humans. PD-1 targeting confers clinical benefits by restoring NK-mediated tumor cell killing in vitro and eliminating tumor cells in vivo in mice engrafted with B-ALL. Conclusions: These results identify PD-1 as a new therapeutic target against leukemic progression, providing new opportunities for the treatment and possibly also the prevention of childhood B-ALL.

## 1. Introduction

In recent years, childhood B-lineage acute lymphoblastic leukemias (B-ALLs) have been extensively characterized at the genomic level [1,2,3,4,5]. Today, the vast majority of B-ALL cases can be categorized into distinct genetic subgroups, each associated with specific prognostic features and treatment responses [6,7,8,9,10,11,12,13,14,15,16,17,18,19,20,21]. In general, B-ALL is characterized by a very low mutational burden [2,5]. Furthermore, in 35–50% of all B-ALL cases, de novo genetic alterations involving genes encoding B-cell transcription factors serve as the primary oncogenic event, determining the disease’s biological and clinical attributes [22]. Similarly, pathogenic germline variants in genes encoding these same transcription factors, such as *PAX5*, predispose to B-ALL development [17,23,24]. However, these germline mutations, whether de novo or inherited, are not sufficient to trigger malignant transformations and require additional oncogenic secondary lesions to induce leukemia formation [14,17,20,22,25,26,27]. Notably, these secondary genetic events frequently also involve B-cell transcription factors and exhibit a remarkable specificity and recurrence pattern relative to the primary lesions.

Notably, alterations in *PAX5,* the gene encoding an essential protein required for B-cell development, are the most common central event in the process of B-cell leukemogenesis in children [1,14,17,20,22,28]. These *PAX5* alterations, including focal deletions, single nucleotide variants and intragenic amplification, have been postulated to arrest lymphoid maturation, a feature characteristic of this disease. Nevertheless, their precise role in B-ALL development is not fully understood.

In this regard, it is well known that a decreased dosage of *Pax5* activity significantly accelerates the development of precursor B-ALL in mice carrying additional fusion genes such as *BCR::ABLp190* and *ETV6::RUNX1* [29,30]. These findings suggest that the secondary *PAX5* genetic alterations that accumulate during the process of malignant transformation might favor an escape from immune surveillance as B-ALL develops. Therefore, reinstating *PAX5* function could potentially serve as a therapeutic approach to halt disease progression and even eliminate leukemic cells. Indeed, experimental evidence has corroborated this hypothesis, demonstrating that reintroducing endogenous *Pax5* expression in *Pax5*-deficient leukemic B cells induces disease remission in murine models [31]. The initiation of *Pax5* expression mediates B-cell commitment during normal hematopoietic differentiation [32], and its removal appears to be required to promote B-ALL development [33]. Although these results suggest that *PAX5* downregulation plays a role in facilitating B-leukemogenesis, such an activity has yet to be directly demonstrated. Determining the mechanisms underlying leukemia formation is essential for improving B-ALL therapy and guiding approaches towards prevention [34,35].

The earliest stages of B-ALL development in children typically go undetected [34], making it nearly impossible to study the initial phases of leukemic transformation in humans. Thus, we have utilized the *Pax5^+/−^* leukemia-prone model to investigate the mechanisms underlying immune evasion during progression to B-ALL. Mice heterozygous for *Pax5*, when exposed to infections, recapitulate the preleukemia-to-leukemia progression found in humans harboring the heterozygous germline *PAX5* c.547G>A pathogenic variant [24,36,37,38,39]. This preleukemia is defined as an at-risk population of normal-appearing cells from which B-ALL develops. These cells have the capacity to undergo malignant transformation; however, leukemia only appears in a small fraction of predisposed carriers when they are exposed to certain environmental factors [34,40].

A substantial body of evidence underscores the significance of inhibitory receptors, commonly referred to as “immune checkpoints,” in tumor-mediated immune suppression. Among immune checkpoint proteins, Programmed Cell Death 1 (PD-1) is mainly expressed in activated T cells and macrophages, and, upon interaction with its ligand PD-L1 (Programmed Cell Death Ligand 1) expressed in tumor cells, PD-1 attenuates antitumor immune responses [41,42,43,44]. In this study, we show that B-ALL progression is associated with the upregulation of PD-1 in preleukemic cells and correlated with the time to malignant transformation in mice. This increase in PD-1 expression is apparent across diverse molecular B-ALL subtypes in mice and humans. Notably, targeting this B-ALL-associated PD-1 expression conferred clinical benefits by restoring NK-mediated tumor cell killing [45] in vitro and eliminating tumor cells in vivo in mice engrafted with B-ALL. These results identify PD-1 as a new therapeutic target in leukemic progression and provide new opportunities for the treatment or even prevention of childhood B-ALL.

## 2. Methods Details

### 2.1. Mouse Model for Natural Immune Stress-Driven Leukemia

*Pax5^+/−^*, *Sca1-ETV6-RUNX1*, *Mb1-Cre* and *Pdcd1^fl/fl^* (*C57BL/6-Pdcd1^tm1^.^1Mrl^*; Taconic Model #13976) mice are as previously described [46,47,48]. *Pax5^+/−^* mice were crossed with *Pdcd1^fl/fl^* mice and *Mb1-Cre* animals to generate mice of the *Pdcd1^fl/fl^*;*Mb1-Cre;Pax5^+/−^* genotype. Mice were bred and maintained under specific pathogen-free (SPF) conditions until exposed to conventional pathogens present in non-SPF animal facilities, as previously described [36,37,38,39,47,49]. *Pdcd1^fl/fl^*;*Mb1-Cre;Pax5^+/−^* mice were treated with a cocktail of antibiotics (ampicillin, 1 g/L, Ratiopharm; vancomycin, 500 mg/L, Cell Pharm; ciprofloxacin, 200 mg/L, Bayer Vital; imipenem, 250 mg/L, MSD; metronidazole, 1 g/L, Fresenius) added to their drinking water ad libitum for a period of eight weeks, with the aim to facilitate leukemia development as previously described [37].

All mouse experiments were performed in accordance with the applicable Spanish and European legal regulations and had been previously authorized by the pertinent institutional committees of both the University of Salamanca and Spanish Research Council (CSIC). Both male and female mice were used in all the studies. Housing environmental conditions included a temperature of 21 °C ± 2 °C, humidity of 55% ± 10% and a 12 h:12 h light/dark cycle. Mice had access to food and water ad libitum. During housing, animals were monitored daily for health status. No data were excluded from the analyses. The inclusion criterion was based on the genotype of the mouse: transgenic versus control littermate. The investigator was not blinded to the group allocation.

*Pax5^+/−^*, *Sca1-ETV6-RUNX1* and *Pdcd1^fl/fl^*;*Mb1-Cre;Pax5^+/−^* mice of a mixed C57BL/6×CBA background were used in this study. For the relevant experiments, *Pax5^+/−^, Sca1-ETV6-RUNX1* and *Pdcd1^fl/fl^;Mb1-Cre;Pax5^+/−^* littermates were used. When animals showed signs of distress, they were humanely euthanized and their organs were harvested. All organs were macroscopically inspected under the stereomicroscope, and representative tissue samples were cut and immediately fixed and stained for subsequent histological analysis. Differences in Kaplan–Meier survival plots of transgenic and WT mice were analyzed using the log-rank (Mantel–Cox) test. Statistical analyses were performed by using GraphPad Prism v8.2.1 (GraphPad Software).

The B-ALL-specific survival curve of *Pdcd1^fl/fl^;Mb1-Cre;Pax5^+/−^* mice (n = 28), *Pax5^+/−^* mice (n = 27) and *Pdcd1^fl/fl^;Mb1-Cre* mice (n = 16) following exposure to common mouse pathogens was defined by the log-rank (Mantel–Cox) test. *p*-values are shown in the corresponding figures.

### 2.2. B-ALL Patient-Derived Xenograft (PDX) Models

Leukemic patient-derived xenografts (PDXs) were obtained from the St. Jude Children’s Research Hospital Public Resource of Patient-derived and Expanded Leukemias (PROPEL) (include https://propel.stjude.cloud (accessed on 22 February 2024)). Briefly, PDXs were established by the tail vein injection of primary human leukemia cells into 8–12-week-old sub-lethally irradiated (250 Rad) NOD.Cg-Prkdc^scid^ IL2rg^tm1Wjl^/SzJ (NSG) mice (The Jackson Laboratory_ RRID:IMSR_JAX:005557) [50]. Spleen cells harvested from engrafted mice were used for expansion in subsequent passages. The level of engraftment was monitored by monthly saphenous vein bleeds and flow cytometric analysis for human CD45^+^ and CD19^+^ cells.

### 2.3. Leukemic Pax5^+/−^ Pro-B-Cell Culture

Iscove’s Modified Dulbecco’s Medium (IMDM- Thermo Fisher Scientific Inc. Alcobendas (Madrid), Spain) supplemented with 50 μmol/L β-mercaptoethanol, 1 mmol/L l-glutamine, 2% heat-inactivated FCS, 1 mmol/L penicillin–streptomycin (BioWhittaker- Merck Life Science S.L.U. Madrid 28006, Spain) and 0.03% (*w*/*v*) primatone RL (Sigma_ Merck Life Science S.L.U. Madrid 28006, Spain) was used for pro-B-cell culture experiments. Leukemic cells isolated by magnetic-activated cell sorting for B220^+^ (Milteny Biotec S.L. 28223 Pozuelo de Alarcón (Madrid) Spain) from BM were cultured on Mitomycin C-treated ST2 cells in IMDM without IL-7 (R&D Systems Bio-Techne R&D Systems, S.L.U. 28046 Madrid, Spain). The ST2 murine cell line, which is derived from mouse bone marrow and is used in research to promote the growth of haematopoietic cells, was obtained from Dr. Meinrad Busslinger [33].

### 2.4. Anti-PD1 Inhibitor In Vitro Experiments

Leukemic *Pax5^+/−^* pro-B cells expressing PD-1 were seeded at 10^6^/3 mL/well in 6-well plates and treated with or without (vehicle) anti-PD1 monoclonal antibody (BioXCell; BE0146; 7.36 μg/mL) for 4, 8 or 24 h before being subjected to a cell viability assay, as described below. Leukemic *Pax5^+/−^* pro-B cells expressing PD-1 were also treated with or without (vehicle) anti-PD1, which lacks the ability to bind to Fc receptors (Fc-silent RMP1-14) for 8 h before being subjected to a cell viability assay.

### 2.5. Viability Assays

Cells from anti-PD1 inhibitor cultures were washed twice with PBS (5′, 1000 rpm, 4 °C). Then, samples were stained with the fixable Zombie NIR viability dye kit (BioLegend; 423105_ Palex Medical S.A. Barcelona 08174-Spain) following the manufacturer’s instructions. Subsequently, cells were washed with PBS supplemented with 1% FCS to eliminate the excess of the viability marker (5′, 1400 rpm, RT). Samples were acquired in a Cytek Northern Light 2000 spectral cytometer (two lasers, red and blue) and analyzed with FCS Express 7 Plus software (version 7.08.0018). The specific fluorescence of the fluorophores and the known forward and orthogonal light scattering properties of mouse cells were used to establish gates. A total of 100,000 cells per sample were assessed. The difference between experimental variables was determined using the Mann–Whitney U test.

### 2.6. BM Transplantation Experiments

Leukemic *Pax5^+/−^* pro-B cells expressing PD-1 were injected intravenously into sub-lethally irradiated (4 Gy) secondary recipient 8-week-old male syngeneic mice (C57BL/6 × CBA), NOD/SCID or nu/nu mice, respectively. Disease development in the recipient mice was monitored by periodic peripheral blood (PB) analysis until blast cells were detected. Then, mice were treated with anti-PD1 or placebo and assessed for B-ALL progression as indicated below. The study of PD-1 targeting as a potential therapeutic approach for childhood B-ALL was performed by using mouse PD-1^+^ leukemic pro-B cells injected through the tail vein of sub-lethally irradiated (4Gy) wild-type (n = 6) mice (C57BL/6 × CBA), NOD/SCID mice (n = 8) or nu/nu (n = 6) mice, respectively. No significant differences were observed in the decrease in the percentage of malignant cells between treated and untreated mice (Mann–Whitney test ns; *p* = 0.4000), showing the inefficacy of the anti-PD1 treatment in the absence of T and NK cells.

### 2.7. Preclinical Therapeutics

Anti-PD1 monoclonal antibody (BioXCell; BE0146_ Abyntek Biopharma, S.L. 48160 Derio—Bizkaia (Spain)) prepared in PBS was administered at 5 μg per mouse, intraperitoneally every 48 h, after animals had developed a detectable leukemic burden (percentage of blasts superior to 15% in PB), as documented by FACS analysis of PB. Disease progression in recipient mice was monitored by periodic PB analysis.

For a detailed description of methods, see the Appendix A.

## 3. Results

### 3.1. B-ALL Development Screens in Genetically Predisposed Mice Identify Cancer Cell-Autonomous Upregulation of the Inhibitory Molecule PD-1

In order to understand the sequential events leading to immune escape during B-ALL development, we used serial bone marrow aspirates, coupled with a sample analysis using spectral flow cytometry, in *Pax5^+/−^* mice to characterize the immuno-regulatory events underlying the preleukemia-to-leukemia conversion (Figure 1). In human patients carrying inherited mutations of *PAX5*, preleukemic cells progress to B-ALL by losing the wild-type *PAX5* allele due to a secondary structural aberration of chromosome 9p [24]. This is phenocopied in the B-ALLs developing in the *Pax5^+/−^* mouse model [36,37,38,39,49], thus demonstrating the importance of the biallelic alterations of *PAX5* for leukemia progression in this B-ALL subgroup. In these mice, upon the loss or marked reduction in *Pax5* activity, the majority of B-ALLs lose *CD19* expression, similar to what is observed in human *PAX5*-associated predisposition to childhood B-ALL [24]. Thus, surface CD19 expression is a very useful surrogate marker of *Pax5* genetic status in mice and allows for the identification of potential immuno-regulatory factors that might correlate with leukemic conversion.

Using the aforementioned approaches, we first analyzed the expression of PD-1 (CD279) and PD-L1 (CD274) on *Pax5^+/−^* preleukemic cells as they switched to a leukemic state following immune stress in the *Pax5^+/−^* model [36,37,38,39,49]. *Pax5^+/−^* preleukemic cells (B220^+^ IgM^−^) did not express PD-1 or PD-L1, and, in spite of their accumulation over time, they were retained in the bone marrow (BM) (Figure 1A, Appendix A). In contrast, PD-1 was upregulated upon the conversion to leukemic B cells in the BM, defined by the lack of CD19 expression in parallel with their emergence in the peripheral blood (PB) (Figure 1A and Appendix A), indicating a loss of or marked reduction in *Pax5* activity similar to what is found in human leukemia [24,36,37,38,39,49]. To examine whether a similar increase in PD-1 expression with leukemia evolution is observed in other mouse leukemia models, we evaluated, together with B-ALLs appearing in *Pax5^+/−^* (n = 14) animals, those that originated in transgenic *Sca1-ETV6-RUNX1* (n = 3) mice [36,47]. We found a detectable PD-1 expression in leukemic cells in 9 of the 14 (64%) *Pax5^+/−^* B-ALL samples, and also in 2 of the 3 (67%) *Sca1-ETV6-RUNX1^+^* B-ALL samples (Figure 1B and Appendix A). All these mouse B-ALLs lacked surface CD19 expression, in agreement with the loss of or marked reduction in *Pax5* activity, similar to what is found in human leukemia [24,36,37,38,39,49]. Thus, we next performed whole exome sequencing (WES) of paired tumor and germline samples from *Pax5^+/−^* tumors to study the status of the wild-type *Pax5* allele within PD-1+ leukemic cells (Appendix A). WES identified four mutations at the *Pax5* locus, as well as other leukemia hotspot mutations such as *Nras*:p.Q61H and *Jak3*:p.R653H. Of note, we could not assess the copy-number or structural alterations via WES. Recurrent genetic alterations affecting the wild-type *Pax5* allele were detected within PD-1+ leukemic cells in some cases (Appendix A). Importantly, no PD-1 expression was detected in normal precursor B cells or in CD19-expressing B-ALLs in either mouse model (Figure 1A,B), consistent with the genetic evidence showing that PD-1 expression is below detection in the presence of *Pax5* [51], and that the expression of PD-1 is consistently upregulated in *Pax5*-deficient precursor B cells [51]. Taken together, these data illustrate that the genetic alterations affecting *Pax5* that accumulate during the process of malignant transformation are associated with PD-1 upregulation upon leukemic conversion in genetically predisposed mice with acquired or germline alterations predisposing to B-ALL.

### 3.2. PD-1 Expression in Human B-ALL

To explore the clinical relevance of these results in humans, we subsequently investigated the expression of PD-1 in primary and xenografted human B-ALL samples, including those of the *PAX5-alt* subtype (where CD19 expression is maintained because the activation of the *CD19* promoter in human B-ALL cells is not solely dependent on *PAX5* activity [52]) and *ETV6::RUNX1^+^* and hyperdiploid subtypes. PD-1 expression was present in 50–100% of these childhood B-ALL xenografts (n = 11) (75% in *PAX5-alt* B-ALLs (3 out of 4 B-ALLs), 50% in *ETV6::RUNX1^+^* B-ALLs (2 out of 4 B-ALLs) and 100% in hyperdiploid B-ALLs (3 out of 3 B-ALLs)) (Figure 2A, Appendix A) and approximately in 31% of diagnostic (11 of 36 B-ALLs) and in 43% of relapsed (9 of 21 B-ALLs) childhood B-ALL primary samples (Figure 2B and Appendix A). However, it is important to note that, in human leukemias like in mice, PD-1 expression is not solely dependent on *PAX5* activity (Appendix A). In summary, PD-1 upregulation occurs in all molecular B-ALL subgroups tested, although the percentage of PD-1^+^ leukemic cells varied from 2 to 100% per sample. Since, until now, PD-1 antibodies are not usually included in most B-ALL marker and diagnostic panels, further prospective studies in a larger dataset of B-ALL cases are needed to confirm the findings linking leukemia cell genomics and PD-1 expression. Still, our observations are consistent with an “immunoediting” process, whereby intratumoral heterogeneity results in the selection of leukemic clones that can avoid elimination by the immune system and thus leave the bone marrow. These findings suggest that there may be a suppression of immune surveillance across human B-ALL subtypes.

### 3.3. PD-1 Is Not Required for Pax5-Dependent B-Cell Leukemogenesis

PD-1 expression has been recently reported as a marker of leukemic stem cells (LSCs) in T-ALL, and anti-PD1 treatment eliminates such LSCs in a cell-autonomous manner [53]. To investigate the role of PD-1 signaling in *Pax5*-dependent B-ALL, and to ascertain if, also in human B-ALLs, PD-1^+^ cells are functionally required for B-leukemogenesis, we used the PD-1 conditional knockout (*Pdcd1^fl/fl^*) mouse (Figure 3). Conditional *Pdcd1^fl/fl^;Mb1-Cre;Pax5^+/−^* mice were generated, where *Pdcd1* is deleted upon B-lineage commitment at the pro-B-cell stage, and we examined whether *Pdcd1^fl/fl^;Mb1-Cre;Pax5^+/−^* animals were prone to infection-induced B-ALL. *Pdcd1^fl/fl^;Mb1-Cre;Pax5^+/−^* mice were exposed to natural infections, and B-ALL development was monitored. We observed that B-ALL appeared in *Pdcd1^fl/fl^;Mb1-Cre;Pax5^+/−^* mice, where 46% (13 out of 28 mice) developed the disease, closely resembling the incidence, latency and overall survival of *Pax5^+/−^* animals (ns, *p*-value = 0.8336) (Figure 3A). This observation supports the notion that PD-1 is not required for B-leukemogenesis. The characterization of *Pdcd1^fl/fl^;Mb1-Cre;Pax5^+/−^* B-ALLs showed that they are histologically and phenotypically similar to those appearing in *Pax5^+/−^* mice (Figure 3B and Appendix A).

To identify somatically acquired second hits leading to B-ALL development in *Pdcd1^fl/fl^;Mb1-Cre;Pax5^+/−^* mice, we performed whole-genome sequencing of paired tumor and germline samples from two *Pdcd1^fl/fl^;Mb1-Cre;Pax5^+/−^* tumors. We identified several somatically acquired recurrent mutations and copy-number variations involving B-cell transcription factors in diseased *Pdcd1^fl/fl^;Mb1-Cre;Pax5^+/−^* mice (Appendix A). To compensate for their inability to upregulate PD-1, the B-ALLs emerging in PD-1-deficient mice express the NK cell inhibitory receptor CD161, something that has never been previously reported in PD-1+ B-ALLs (Figure 3D and Appendix A). Overall, the same genes appear mutated in *Pax5^+/−^* murine leukemias [36,37,38,39,49] as in human B-ALL samples [2,5,17,54]. Thus, the drivers of B-ALL appear similar in *Pdcd1^fl/fl^;Mb1-Cre;Pax5^+/−^* and *Pax5^+/−^* mouse leukemic cells as well as human B-ALL blasts. However, in the absence of PD-1, tumor cells express the NK cell inhibitory receptor CD161, highlighting the importance of NK-mediated immune surveillance and the necessity for leukemic cells to evade the host’s NK immune response in order to exit the bone marrow. Future studies exploring cohorts of patients are required to better establish whether CD161 expression implies a new category of immunoevasive B-ALL subtypes.

### 3.4. Anti-PD1 Treatment Restores the Immune Capacity to Eliminate PD-1-Positive Tumor Cells in Mice Engrafted with B-ALL

Building on our findings, we next determined the biological consequences of PD-1 upregulation on antitumor immune response. Thus, we examined the potential of targeting PD-1 for B-ALL treatment (Figure 4). Here, in order to better understand the results from spectral cytometry, it is important to mention that, since one of the main functions of Pax5 is maintaining non-B-cell genes repressed during B-cell differentiation, B-ALL tumors arising in *Pax5^+/−^* mice may mimic a *Pax5^−/−^* phenotype and express promiscuous surface lineage markers such as CD8, Mac1 or Gr1 (Figure 4B). In contrast to what has been reported for T-ALL PD-1^+^ leukemia stem cells [53], in vitro PD-1 targeting did not induce the apoptosis of PD-1^+^ B-ALL cells (either of human or mouse origin) (Appendix A). However, in vivo PD-1 targeting in B-ALL mice efficiently reduced disease burden and extended overall survival versus placebo-treated mice (*p* = 0.0177; Figure 4A–C and Appendix A). These findings suggest that PD-1 targeting released the immune-suppressive regulation and restored the tumor-specific cytotoxicity of either cytotoxic T lymphocytes or NK cells, which are the cellular components mediating the effects of the PD-1 blockade [55,56].

### 3.5. PD-1 Targeting Sensitizes PD-1-Positive B-ALLs to NK Cell-Mediated Killing

To determine whether the therapeutic effect of PD-1 targeting in B-ALL mice is dependent on T or NK cells, we repeated the checkpoint targeting strategy (Figure 4) in NOD/SCID mice, which lack B and T cells and have defective natural killer (NK) cell function. We recapitulated B-ALL disease in NOD/SCID mice by tail-vein injecting the PD-1^+^ leukemic B cells. To dissect the impact of PD-1 targeting in leukemic NOD/SCID mice, we initiated the treatment once blast cells were detected in PB. Unlike in immunocompetent animals, PD-1 targeting in PD-1^+^ B-ALL NOD/SCID mice did not reduce disease burden (Figure 4D). This similar in vivo growth of tumor cells in NOD/SCID mice excluded a tumor cell-intrinsic effect of PD-1-targeting antibodies. However, when we similarly recapitulated PD-1^+^ B-ALL disease in nu/nu mice, which lack T cells but possess NK cells, PD-1 targeting efficiently reduced the disease burden (Figure 4E,F). These findings are consistent with previous observations suggesting a T-cell independent immunosurveillance mechanism in the conversion of the preleukemic clone into a full-blown B-ALL [34,37,57]. To better understand how B-ALLs can escape NK cell surveillance through the PD-1 checkpoint, we performed ex vivo NK cytotoxicity assays using both mouse (S748 cells) and human B-ALL (REH cells). NK cytotoxicity assays showed that the targeting of PD-1 using the specific anti-PD1 antibody sensitizes both mouse and human B-ALLs to NK cell-mediated killing (Figure 5A–E). Taken together, our results indicate that PD-1 is central to mediating the in vivo escape of B-ALL cells from surveillance by NK cells, and that PD-1 targeting conferred clinical benefits by restoring NK-mediated tumor cell killing in vitro and eliminating tumor cells in vivo in mice engrafted with B-ALL.

### 3.6. PD-1 Directly Inhibits the Antitumor Activity of NK Cells on B-ALL

It has been proposed that the acquisition of PD-1 in the membrane via trogocytosis can inhibit NK cell response against tumor cells expressing PD-1 [58]. To test this hypothesis, we initially used PD-1^+^ S748 cells, which derive from the transformation of murine *Pax5^+/−^* B cells under immune stress [36]; these S748 cells express high levels of PD-1 (Figure 5A), and we have used them extensively in previous studies [36]. We now co-cultured PD-1^+^ S748 leukemic cells together with NK cells. In the absence of tumor cells, NK cells did not stain for PD-1 (Figure 5F). Similarly, NK cells did not stain positively for PD-1 when incubated with PD-1^+^ S748 leukemic cells (Figure 5F), indicating that PD-1 was not endogenously expressed by NK cells and was not acquired from leukemic cells in these settings. Likewise, co-culturing preleukemic (i.e., PD-1-negative) *Pax5^+/−^* cells with NK cells leads to cytotoxicity that is not affected by treatment with anti-PD-1 antibodies (Appendix A). Finally, to examine whether the therapeutic effect of PD-1 antibodies was due to antibody-dependent cellular cytotoxicity (ADCC), likely mediated by NK cells against PD-1^+^ leukemic cells coated with anti-PD1 antibodies, we used an engineered version of anti-PD-1 that lacks the ability to bind to Fc receptors (Fc-silent RMP1-14) [59]. A treatment with Fc-silent anti-PD1 antibodies did not affect the growth of PD-1^+^ S748 leukemic cells (Figure 5G,H), indicating that the therapeutic effect of anti-PD-1 antibodies was due to ADCC. Together, these results indicate that PD-1 expression by leukemic cells suppresses NK cell-mediated antitumor activity, facilitating immune evasion, bone marrow exit and disease dissemination. Importantly, this process can be prevented by treatment with PD-1-targeting antibodies. However, it cannot be discarded that there could be another mechanistic explanation for the protective function of PD-1 on leukemia cells, like, for example, the modulation of activating/inhibitory NK receptors.

## 4. Discussion

If childhood B-cell acute lymphoblastic leukemia (B-ALL) happens only in genetically predisposed individuals and is triggered by immune stress, then it is very likely a preventable disease. Therefore, it becomes imperative to delineate the processes implicated in leukemic progression, as these processes may provide windows of opportunity to intervene by stopping B-ALL in its tracks.

Here, our studies have uncovered a previously unknown molecular process that takes place during the initial stages of childhood B-cell leukemogenesis, allowing us to discover a readout of *PAX5* inactivation, the most frequently altered transcription factor in B-ALL [1,14,17,20,22,28]. We demonstrate that the *Pax5* genetic alterations that accumulate during the process of malignant transformation are associated with PD-1 upregulation upon the time of conversion to B-ALL in mice. PD-1 is upregulated in murine Pax5-deficient B-ALL because PD-1 is a target gene repressed by PAX5 in normal conditions [51]. This increase in PD-1 expression is also apparent across diverse molecular B-ALL subtypes in human B-ALL samples, although in mouse and human leukemias PD-1 expression is not solely dependent on PAX5 activity. However, it is not known whether PD-1 upregulation in Pax5-deficient B-ALL is accompanied by changes in membrane microdomain organization. This increase in PD-1 expression is also apparent across diverse molecular B-ALL subtypes in human B-ALL samples, although in mouse and human leukemias PD-1 expression is not solely dependent on PAX5 activity, and further studies should help to identify the mechanisms involved. In addition, PD-1 upregulation protects B-ALL blasts from NK-mediated immune surveillance. Conversely to what was observed in T-ALL [53], PD-1 is not functionally required for B-leukemogenesis. However, in the absence of PD-1, leukemic cells upregulate the NK cell inhibitory receptor CD161, underscoring the critical role of NK cell-mediated immune surveillance. This suggests that the evasion of NK cell-mediated immunity is essential for leukemic cells to exit the bone marrow. The importance of shielding B-ALL blasts from NK surveillance is further reinforced by the upregulation of alternative NK cell inhibitory molecules in B-ALLs arising in PD-1-deficient mice. Finally, our findings reveal that PD-1 targeting can serve as a therapeutic strategy to promote the NK cell-mediated killing of leukemia cells, providing new opportunities for the treatment or prevention of childhood B-ALL. Our findings could also justify the use of small molecular PD-1 inhibitors for oral administration in B-ALL patients with PD-1-expressing leukemic cells, currently in development for some solid tumors [60]. Nevertheless, the use of single agents against cancer is not very prudent and, therefore, combining PD-1 blockades with standard drugs could help in managing B-ALL better. However, this anti-PD-1 therapy could fail in clinical settings through the acquisition of other checkpoint inhibitors, and it can be associated with potential risks, like immunosuppression, autoimmunity, etc., that should be minimized by the early identification and early onset of a prophylactic treatment.

It is important to underscore that the identification of a biomarker correlating with environmental exposures along the trajectory of leukemic transformation has been achieved using predisposed mouse models where the B-ALL disease emerges naturally. This is relevant because many of the observations obtained previously using these mouse models have later been mirrored in pediatric B-ALL patients, as illustrated by the case where the B-cell alterations found in preleukemic *Pax5^+/−^* mice [36] were later confirmed in children carrying PAX5 germline variants [61], or by the discovery that B-ALL driver genes are not targeted by AID in mice, subsequently validated in human ALL blasts [49], or by the identification of gut microbiome immaturity in B-ALL-predisposed mice [37], which was also later corroborated in children with B-ALL [62]. Hence, these preclinical mouse models, where B-ALL occurs naturally, are indispensable for elucidating the early phases of B-ALL development, which are typically unnoticed in children [34], making it nearly impossible to study the initial phases of leukemic transformation in humans. Likewise, recently it has been shown that NK cell-mediated cytotoxicity shapes the clonal evolution of human B-cell leukemia by showing that tumor cells are actively edited by NK cells during the equilibrium phase and use different avenues to escape NK cell-mediated eradication [63].

## 5. Conclusions

Overall, our study demonstrates that certain cases of human B-ALL express PD-1, and that targeting this pathway with checkpoint inhibitors can activate NK cell activity, thereby presenting a potential therapeutic opportunity. In fact, our data show that sustained leukemia suppression can be achieved with anti-PD1 single-agent treatment in the presence of a functional NK compartment. Accordingly, these findings form the foundation for a potential new approach to treat B-ALL, the commonest form of pediatric cancer and the leading cause of cancer-related death in children. These results are also of immediate diagnostic importance because they suggest that most PD-1-positive B-ALL cases may be rapidly detected using flow cytometry immunophenotyping to further guide classification and tailored therapy. Future studies exploring larger cohorts of patients will be required to better establish how PD-1 upregulation may be used to direct the daily clinical care of children with B-ALL.

## Figures and Tables

**Figure 1 hematolrep-17-00061-f001:**
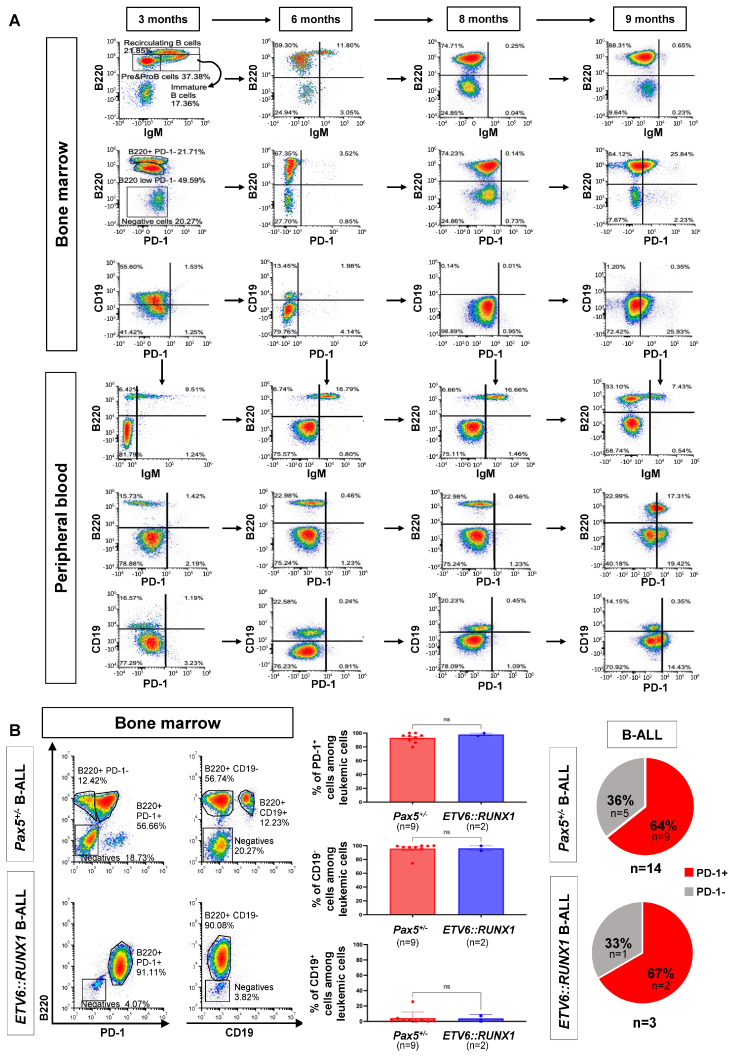
PD-1 expression in B-ALL. (**A**) PD-1 expression can be detected in leukemic cells at the time they appear in peripheral blood. Periodical bone marrow (BM) analysis demonstrated the existence of a premalignant population in the BM of *Pax5^+/−^* mice at the age of 6 months (**upper panels**). At that time, the preleukemic cells (gated as Zombie Nir^−^ CD45^+^) were B220^+^ IgM^−^ PD-1^−^ and retained in the BM. This premalignant population was monitored for three months in the BM and peripheral blood (PB). By the time malignant cells were detectable in the PB (9 months, **rightmost panels**), PD-1 was expressed by the leukemic population. Notably, PD-1 was only expressed when leukemic cells left the BM. (**B**) PD-1 expression is detected in mouse B-ALL of different genetic subtypes. Flow cytometry analysis of one B-ALL *Pax5****^+/−^*** mouse (**upper-left panels**) and one B-ALL *Sca1-ETV6-RUNX1* mouse (**bottom-left panels**) revealed expression of PD-1 in the leukemic population in the BM (gated as PI^−^ B220^+^ PD-1^+^ PD-L1^−^ CD19^−^). In both cases, leukemia cells were B220^+^ CD19^−^ PD-1^+^ PD-L1^−^. Bar graphs (**middle panels**) show the percentage of PD-1^+^, CD19^−^ and CD19^+^ cells among the leukemic population in the *Pax5^+/−^* (n = 9) and ETV6::RUNX1 (n = 2) B-ALL-diseased mice analyzed. Each dot represents an individual mouse, and bars indicate the mean ± SEM; Mann–Whitney test: PD-1^+^ ns *p* = 0.5636, CD19^−^ ns *p* = 0.8727 and CD19^+^ ns *p* = 0.8727. In total, 64% of the *Pax5^+/−^* B-ALL cases (9 out of 14 mice) and 67% of the ETV6::RUNX1^+^ B-ALL cases (2 out of 3 mice) studied for PD-1 expression through spectral flow cytometry expressed the protein among the leukemia cells (right panel). The gating strategy is shown in Appendix A.

**Figure 2 hematolrep-17-00061-f002:**
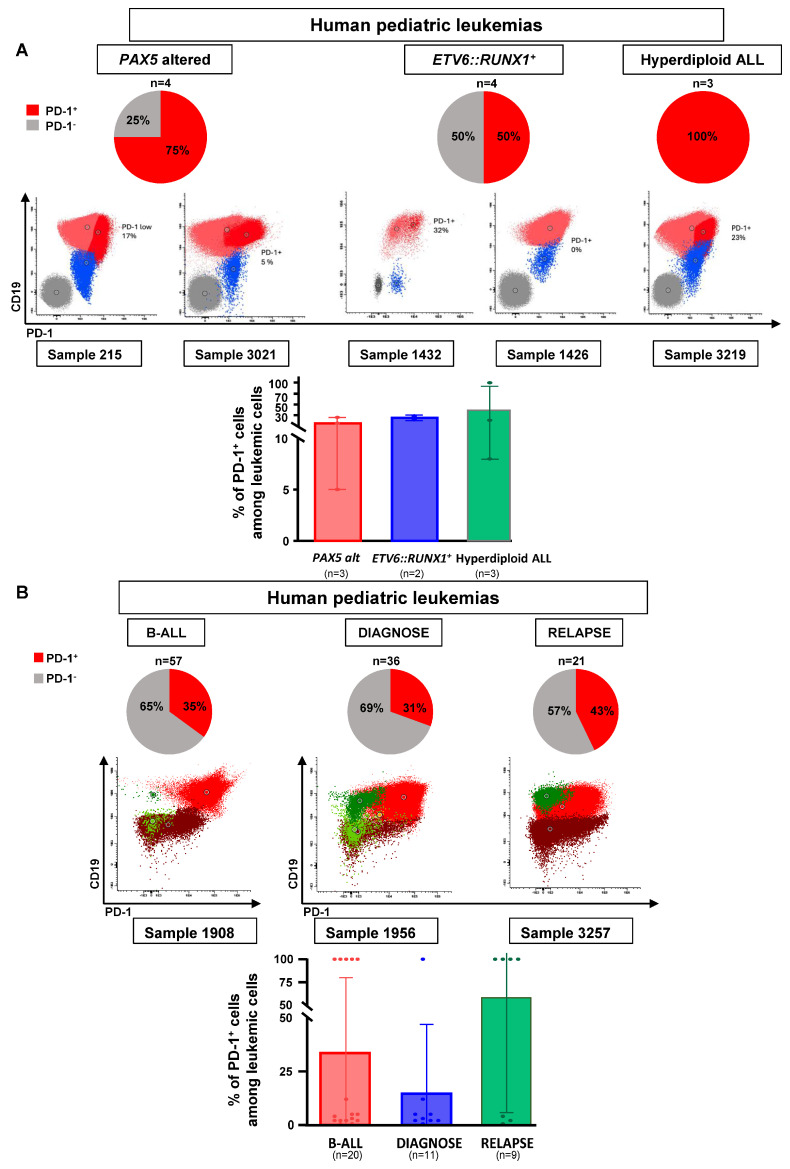
PD-1 expression in human B-ALL. (**A**) Expression of PD-1 in B-ALL PDX models. Analysis of 11 samples from childhood leukemias grown as PDX demonstrated the expression of PD-1 (CD279) in different genetic subtypes. Analysis by spectral flow cytometry of *PAX5-alt* B-ALL (n = 4), *ETV6::RUNX1^+^* B-ALL (n = 4) and hyperdiploid B-ALL (n = 3) confirmed the expression of PD-1 in 3/4 (75%), 2/4 (50%) and 3/3 (100%) of the samples, respectively. Spleens of the recipient mice were used for the isolation and flow cytometry analysis of leukemic cells (gated as CD19^+^ CD5^−^ CD45^+^). Bar graphs show the percentage of PD-1^+^ cells among the leukemic population: each dot represents an individual sample; bars indicate mean ± SEM. Statistical analysis (Mann–Whitney test) revealed no significant differences: *PAX5-alt* vs. *ETV6::RUNX1, p* = 0.248; *PAX5-alt* vs. hyperdiploid, *p* = 0.513; *ETV6::RUNX1* vs. hyperdiploid, *p* = 0.564. Spectral flow cytometric analysis shows the leukemic population in six different examples: PD-1^+^ leukemic population is depicted in dark red and the PD-1^−^ leukemic population is depicted in light red. (**B**) Expression of PD-1 in human pediatric B-ALL. Analysis of 59 cases from BM samples of childhood B-cell leukemias confirmed the expression of PD-1 (CD279) in 35% of the cases. The samples were representative of the different genetic subtypes of the disease. Bar graphs depict the percentage of PD-1^+^ cells among the leukemic population in the total cohort (n = 57), at diagnosis (n = 36) and at relapse (n = 21). Each dot represents an individual sample; bars indicate mean ± SEM. Mann–Whitney test: n.s. (diagnosis vs. relapse, *p* = 0.233). Spectral flow cytometry analysis shows the leukemic population in three different examples: PD-1^+/−^ leukemic population is depicted in red.

**Figure 3 hematolrep-17-00061-f003:**
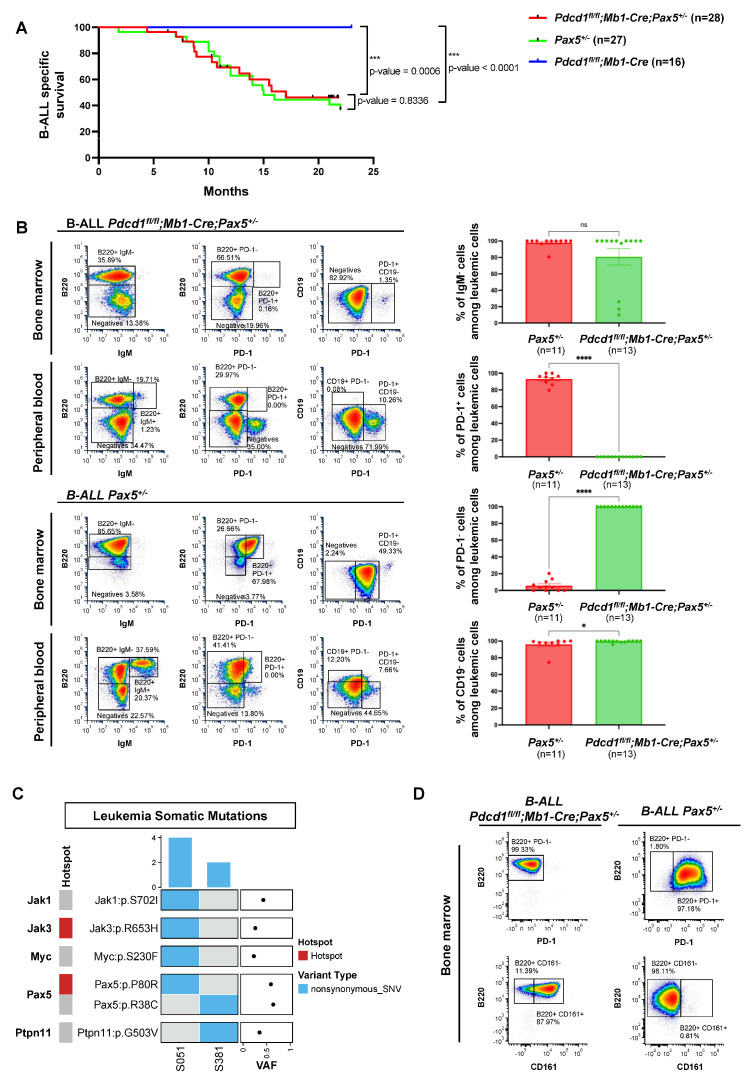
PD-1 is not required for Pax5-dependent B-cell leukemogenesis. (**A**) B-ALL-specific survival curve of *Pdcd1^fl/fl^;Mb1-Cre;Pax5^+/−^* mice (red line, n = 28), *Pax5^+/−^* mice (green line, n = 27) and *Pdcd1^fl/fl^;Mb1-Cre* mice (blue line, n = 16) following exposure to common mouse pathogens. Log-rank (Mantel–Cox) *p*-values are shown. (**B**) Representative spectral flow cytometry plots of the BM and PB of a B-ALL-diseased *Pdcd1^fl/fl^;Mb1-Cre;Pax5^+/−^* mouse (upper-left panels) compared to a B-ALL-diseased *Pax5^+/−^* mouse (lower-left panels) that show the phenotypic similarity of the leukemias, except for the absence of PD-1 expression in leukemic cells in the *Pdcd1^fl/fl^;Mb1-Cre;Pax5^+/−^* mouse. Bar graphs (right panels) show the percentage of IgM^−^, PD-1^+^, PD-1^−^ and CD19^−^ cells among the leukemic population in the *Pax5^+/−^* (n = 9) and *Pdcd1^fl/fl^;Mb1-Cre;Pax5^+/−^* (n = 13) B-ALL-diseased mice analyzed. Each dot represents an individual mouse, and bars indicate the mean ± SEM; Mann–Whitney test: IgM^−^ ns *p* = 0.2774, PD-1^+^ cells *p* < 0.0001, PD-1^−^ cells *p* < 0.0001 and CD19^−^ *p* = 0.0328. (**C**) Whole-genome sequencing in leukemic *Pdcd1^fl/fl^;Mb1-Cre;Pax5^+/−^* mice. Oncoprint of the main somatic single-nucleotide mutations and copy-number alterations across two leukemia samples. Somatic alterations are clustered by gene. Tumor DNA was derived from whole leukemic bone marrow (BM) or lymph nodes (LNs), while tail DNA of the respective mouse was used as reference germline material. The percentage of leukemic cells in the sequenced samples was 73% and 50%, respectively. Previously reported known human or mouse leukemia hotspot mutations are highlighted (red). Mean tumor variant allele fraction (VAF) for each single nucleotide mutation is shown on the dot plot on the right. (**D**) Representative spectral flow cytometry plots of the BM of a B-ALL-diseased *Pdcd1^fl/fl^;Mb1-Cre;Pax5^+/−^* mouse (right panels) compared to a B-ALL-diseased *Pax5^+/−^* mouse (left panels) that show the expression of CD161 in leukemic cells in the absence of PD-1 expression. The gating strategy is shown in Appendix A. (Note: * = *p* < 0.05; *** = *p* < 0.001; **** = *p* < 0.0001; ns = not significant (*p* ≥ 0.05).

**Figure 4 hematolrep-17-00061-f004:**
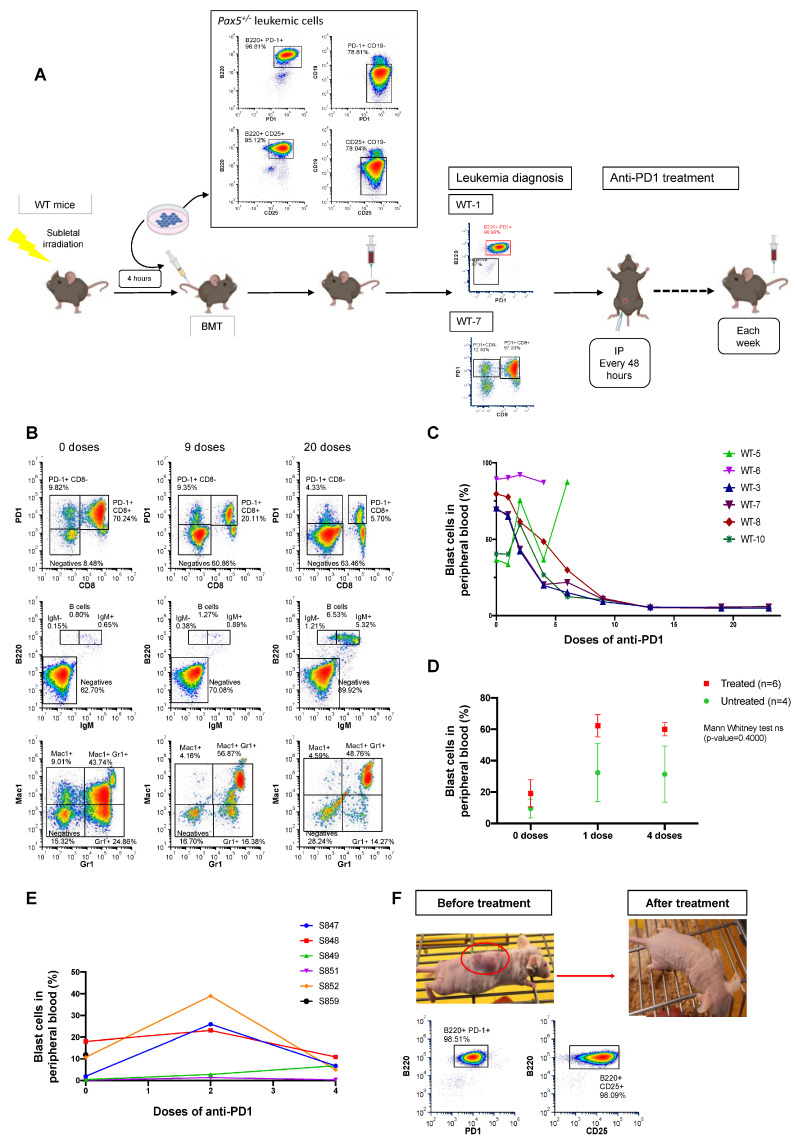
PD-1 targeting is a potential therapeutic approach for childhood B-ALL. (**A**) Experimental design. Mouse PD-1^+^ leukemic pro-B cells were injected through the tail vein of sub-lethally irradiated (4Gy) wild-type (n = 6) mice (C57BL/6 × CBA), NOD/SCID mice (n = 8) or nu/nu (n = 6) mice, respectively. Disease development in the recipient mice was monitored by periodic peripheral blood (PB) analysis until blast cells were detected. Then, mice were treated with either anti-PD1 antibody or placebo and assessed for B-ALL evolution through flow cytometry analysis. Anti-PD1 treatment or placebo was administered intraperitoneally every 48 h. B-ALL progression was monitored weekly through PB flow cytometry analysis. (WT-1 and WT-7 correspond to the mouse codes; BMT: bone marrow transplant). (**B**) Anti-PD1 treatment reduced disease burden and allowed the reconstitution of a healthy immune system. Representative flow cytometry analysis from a mouse treated with several doses of anti-PD1. Please note that, since one of the main functions of Pax5 is keeping non-B-cell genes repressed, B-ALL tumors arising in *Pax5^+/−^* mice may mimic a *Pax5^−/−^* phenotype and express lineage-promiscuous markers such as CD8, Mac1 or Gr1, as shown in this figure. The first column showed the presence of a malignant population (mainly PD-1^+^ CD8^+^, upper-leftmost panel, and Gr1^+^, lower-leftmost panel) in PB. The percentage of leukemia cells was so high that no other healthy cells were detectable in PB (see no IgM^+^ cells in middle-leftmost panel). In the second and third columns, panels show how anti-PD1 treatment efficiently reduced the percentage of these malignant cells in the PB and allowed the reconstitution of the normal hematopoiesis after 20 doses of anti-PD1 antibody. No side effects associated with the use of the anti-PD1 antibody were observed. (**C**) Anti-PD1 treatment significantly reduced the percentage of blast cells in PB of B-ALL-transplanted animals. Repeated administration of anti-PD1 reduced leukemia burden in transplanted WT animals in four of the six mice (66%), with the percentage of malignant cells becoming negligible after 2 months of treatment. No signs of toxicity were observed in the animals treated with more than 20 doses of the drug, and no side effects associated with the use of the anti-PD1 antibody were observed. In two mice (WT-5 and WT-6), the treatment did not show the same success; in the case of WT6 mouse, treatment failure was most likely due to the very high percentage of leukemia cells present in PB by the time the treatment was started; the lack of response of WT5 mouse may reflect the disparity of responses to cancer therapy, something commonly observed in humans, where patients may exhibit different responses to the same treatment. (**D**) Anti-PD1 treatment did not reduce leukemic burden in mice lacking T and NK cells. The same experimental procedure shown in 4A was followed but using NOD/SCID mice (n = 12): 8 of them were treated with anti-PD1 once leukemia was diagnosed and the other 4 were treated with vehicle (labeled as “untreated”). No significant differences were observed in the decrease in the percentage of malignant cells between treated and untreated mice (Mann–Whitney test ns; *p* = 0.4000), showing the inefficacy of the anti-PD1 treatment in the absence of T and NK cells. (**E**) Anti-PD1 treatment efficiently reduced leukemia burden in nu/nu mice. The same experimental procedure shown in Figure 4A was followed, but using nu/nu mice (n = 6). After the administration of four doses of anti-PD1, the percentage of leukemic cells was significantly reduced, therefore showing the efficacy of the anti-PD1 treatment in the presence of endogenous NK cells. Nu/nu mice did show signs of toxicity after several doses of anti-PD1 treatment, and they died due to side effects as a result of the absence of T cells in this mouse model. (**F**) Anti-PD1 treatment eradicated the subcutaneous tumors formed by PD-1^+^ leukemic B cells. Following adoptive transfer of leukemic PD-1^+^ *Pax5^+/−^* B cells, five out of six (83%) nu/nu mice developed subcutaneous nodules due to accumulation of leukemic cells under the skin, as identified by FACS analysis. Animals that developed subcutaneous tumors were treated with the same regimen of anti-PD1 as B-ALL-diseased mice. The subcutaneous tumors disappeared after administration of 7–10 doses of anti-PD1. The gating strategy is shown in Appendix A.

**Figure 5 hematolrep-17-00061-f005:**
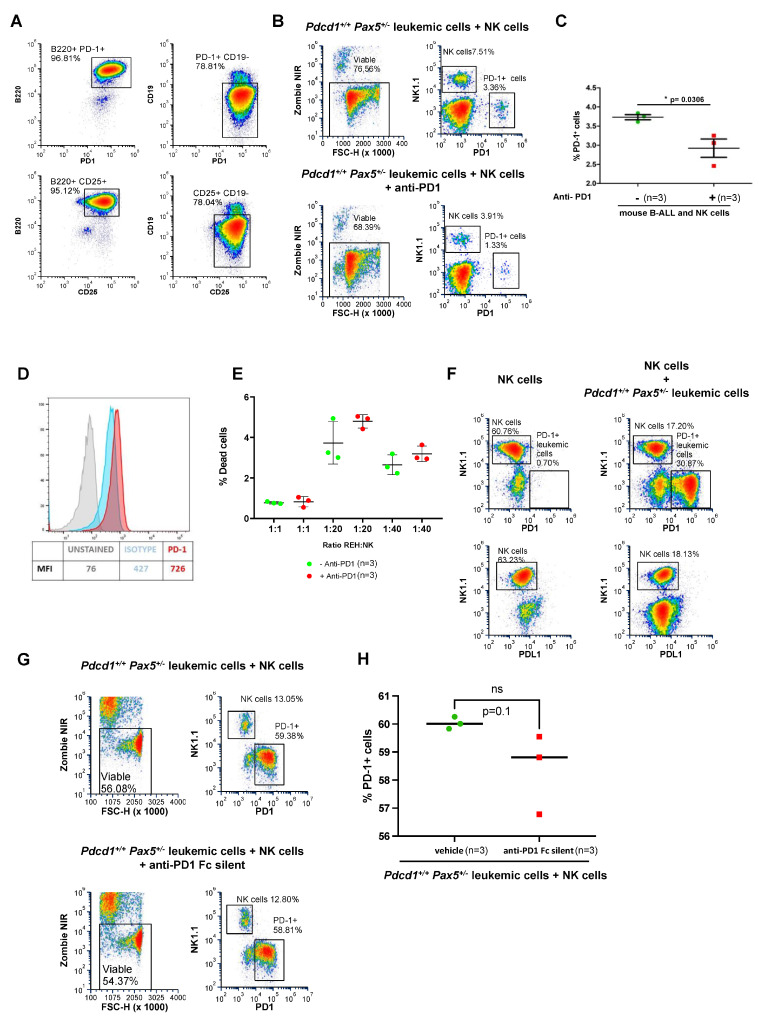
PD-1 targeting sensitizes PD-1^+^ B-ALLs to NK cell-mediated killing. (**A**) Expression of PD-1 in murine *Pdcd1^+/+^* leukemic B cells (S748). Flow cytometry analysis showing the expression of PD-1 in the CD19^−^ *Pax5^+/−^* transformed cells in culture (**upper panels**). The leukemic cells also express CD25, a typical marker of leukemic B cells (**lower panels**). Cells were acquired from Cytek Northern Lights 2000 and data were analyzed with FCS Express software. (**B**) PD-1 targeting sensitizes B-ALLs to NK cell-mediated killing. Murine leukemic *Pax5^+/−^* pro-B cells expressing PD-1 were co-cultured with murine NK cells (1:1) in the presence of anti-PD1 antibody for 8 h (n = 3). The same cells cultured in the same conditions but with vehicle (PBS) were used as a control (n = 3). (**C**) The therapeutic effect of PD-1 antibodies was due to an antibody-dependent cellular cytotoxicity (ADCC) mechanism. Antibody-dependent cellular cytotoxicity (ADCC) using a PD-1 monoclonal antibody was quantitated through flow cytometry. An unpaired t-test was used to detect differences between the two conditions of culture. Error bars represent the mean and standard deviation. (**D**) PD-1 expression in human *ETV6::RUNX1*-positive B-ALL cells (REH cells). Histograms illustrating the pattern of expression of PD-1 in human *PAX5*-mutant REH cells where mean fluorescence intensity (MFI) is indicated. (**E**) PD-1 targeting sensitizes human B-ALLs to NK cell-mediated killing. Human REH cells were co-cultured with human NK cells at different effector/target ratios in the presence (red) or absence (green) of anti-PD1 for 8 h (n = 3). Antibody-dependent cellular cytotoxicity (ADCC) was quantitated by flow cytometry. Mean and mean standard errors are shown. Wilcoxon test *p*-value = 0.05. (**F**) Expression of PD-1 and PD-L1 in NK cells. NK cells did not stain for PD-1 or PD-L1 when either cultured alone (**left panels**) or after being co-cultured with PD-1^+^ leukemic B cells for 8 h (**right panels**). (**G**) PD-1 targeting using an Fc-silent antibody that lacks the ability to bind to Fc receptors does not sensitize B-ALL cells to NK cell-mediated killing. Murine leukemic *Pax5^+/−^* pro-B cells expressing PD-1 were co-cultured with murine NK cells (1:1) in the presence of an anti-PD1 monoclonal antibody that lacks the ability to bind to Fc receptors (Fc-silent RMP1-14) for 8 h (n = 3). The same cells cultured in the same conditions but with vehicle (PBS) were used as a control (n = 3). (**H**) Antibody-dependent cellular cytotoxicity using an Fc-silent PD-1 monoclonal antibody. The ADCC using an Fc-silent PD-1 monoclonal antibody was quantitated through flow cytometry. An unpaired t-test was used to evaluate the differences between the two conditions. The gating strategy is shown in Appendix A.

## Data Availability

The data presented in this study are openly available in to NCBI’s Sequence Read Archive (SRA) under the BioProject accession number PRJNA1084753. This paper does not report original code. Preprints ID: https://doi.org/10.20944/preprints202505.0386.v1.

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
