# Peer review of "PD-1 Expression Promotes Immune Evasion in B-ALL"

_hematolrep, 2025, doi:10.3390/hematolrep17060061_

Round 1
Reviewer 1 Report
This study demonstrated that inherited or germline predispositions ( PAX5 mutations in this case) is associated with PD-1 up regulation in B-ALL cells, which could cause select clones to evade immune surveillance and contribute to disease evolution, therapy resistance and relapse. Furthermore, the authors propose assessing PD-1 levels to identify a patient cohort who might benefit from blocking PD-1 therapeutically.
Following are the major concerns to be addressed:
- It is well understood the relation between Pax-5 alteration and PD-1 up regulation, however what triggers PD-1 up regulation in other genetic background like ETV:RUNX1 or hyperploid B-ALL is not clear and should be explained.
- The density plots for flow cytometry in Figure 1A, 1B, 3B, should be accompanied by bar graphs or equivalent. This would help to figure out the variability and extent of PD-1 expression/positivity in B-ALL samples.
- Across the Results section, percentages and frequencies (e.g., “64% of Pax5+/− B-ALLs expressed PD-1”) have been reported casually. Adding statistics to these statements like using binomial confidence intervals ( eg- Wilson) will enforce rigor. Or, include numerator/denominator like "9 of 14 tumors (64%) expressed PD-1".
- Results section 3.3-The Pdcd1fl/fl;Mb1-Cre;Pax5+/− data is intriguing in showing that PD-1 is not required for B-ALL. Especially, the data related to CD161 expression as alternative immune evasion strategies should a part of the main figure and discussed in sufficient detail.
- Figure4 F; a measurement of tumor volume as a bar graph should be included if possible.
- Data from B-ALL human cohort (cbioportal) should be utilized to assess PD-1 expression vs overall survival, response to treatment. This will further bolster the findings of this paper in patient samples.
Other comments:
Single agents against cancer is not very prudent. Combining PD-1 blockade with standard drugs could help in managing B-ALL better, and be included in discussion.
Also, discuss potential mechanism by which proposed anti PD-1 therapy could fail in clinical settings like acquisition of other checkpoint inhibitors.
Author Response
See pdf attached

Reviewer 2 Report
The manuscript presents the results on the role of PD-1 in B-ALL. It is an interesting work, however, there are several major issues:
-
In the Methods section, the authors should provide detailed information on all reagents and antibodies used, including catalog numbers and suppliers.
-
It is also unclear how pro-B cells were isolated, which markers where used? It is also unclear whether the authors used the B220+ kit from Miltenyi for positive selection, which may activate B cells during the process of isolation.
2. All flow cytometry results are currently presented as representative plots. The authors should include quantitative graphs summarizing data from all replicates. Gating strategy should also be provided.
- Methods section. Authors should include separate sections called "Statistics" and add the information about statistical analyses used.
- Methods section: Authors should include section on flow cytometry, adding information on staining, antibodies used, etc.
- Results section: Authors should include figures summarizing data for all replicates, with significance levels.
Author Response
See pdf attached

Reviewer 3 Report
The manuscript of Ana Casado-Garcia et al investigates how preleukemic B cells acquire resistance to immune surveillance, focusing on the role of PD-1 upregulation upon Pax5 inactivation. The study combines murine models, patient-derived xenografts, and human B-ALL samples, providing strong translational value. The evidence that PD-1 targeting restores NK-mediated tumor cell killing is particularly compelling.
However, some aspects require clarification:
-
The mechanistic link between Pax5 loss and PD-1 upregulation is not fully elucidated. A more detailed discussion of possible transcriptional or epigenetic regulators would improve the manuscript.
-
The observation that PD-1-deficient leukemias compensate by expressing CD161 is intriguing. Do human B-ALL cases show similar compensatory pathways?
-
Therapeutic implications of PD-1 blockade appear largely NK cell–dependent. The authors should expand on the relevance of this mechanism in pediatric patients.
-
Please discuss whether PD-1 upregulation in Pax5-deficient B-ALL is accompanied by changes in membrane microdomain organization. PD-1 nanocluster formation and partitioning into cholesterol-rich lipid rafts can modulate receptor signaling and NK-cell synapse function
The manuscript of Ana Casado-Garcia et al investigates how preleukemic B cells acquire resistance to immune surveillance, focusing on the role of PD-1 upregulation upon Pax5 inactivation. The study combines murine models, patient-derived xenografts, and human B-ALL samples, providing strong translational value. The evidence that PD-1 targeting restores NK-mediated tumor cell killing is particularly compelling.
However, some aspects require clarification:
-
The mechanistic link between Pax5 loss and PD-1 upregulation is not fully elucidated. A more detailed discussion of possible transcriptional or epigenetic regulators would improve the manuscript.
-
The observation that PD-1-deficient leukemias compensate by expressing CD161 is intriguing. Do human B-ALL cases show similar compensatory pathways?
-
Therapeutic implications of PD-1 blockade appear largely NK cell–dependent. The authors should expand on the relevance of this mechanism in pediatric patients.
-
Please discuss whether PD-1 upregulation in Pax5-deficient B-ALL is accompanied by changes in membrane microdomain organization. PD-1 nanocluster formation and partitioning into cholesterol-rich lipid rafts can modulate receptor signaling and NK-cell synapse function
Author Response
See pdf attached

Round 2
Reviewer 2 Report
The authors should revise the manuscript, focusing on appropriate presentation of flow data.
Flow cytometry data presentation require extensive revision. Please provide graphs/bar charts, summarising all experiments. The Figures should be included in the main Result section along with the representative plots.
Author Response
See pdf attached

Reviewer 3 Report
The authors have completed my requests
The authors have completed my requests
Author Response
Thank you very much